# General practitioners' perceptions of delayed antibiotic prescription for respiratory tract infections: A phenomenographic study

**Erika A. Saliba-Gustafsson**[1]*, **Marta Röing**[2], **Michael A. Borg**[3,4], **Senia Rosales-Klintz**[1,5], **Cecilia Stålsby Lundborg**[1]

1 Department of Global Public Health, Health Systems and Policy: Improving Use of Medicines, Karolinska Institutet, Stockholm, Sweden, 2 Department of Public Health and Caring Sciences, Health Services Research, Uppsala University, Uppsala, Sweden, 3 Department of Infection Prevention and Control, Mater Dei Hospital, Msida, Malta, 4 Faculty of Medicine and Surgery, University of Malta, Msida, Malta, 5 Unit of Surveillance and Response Support, European Centre for Disease Prevention and Control, Solna, Sweden

* erika.saliba@gmail.com

**Data Availability Statement:** We note your current Data Availability Statement states "Since we present interview data, data cannot be shared

## Abstract

### Background

Antibiotic use is a major driver of antibiotic resistance. Although delayed antibiotic prescription is a recommended strategy to reduce antibiotic use, practices vary; it appears less commonly used in southern European countries where antibiotic consumption is highest. Despite these variations, few qualitative studies have explored general practitioners' perceptions of delayed antibiotic prescription. We therefore aimed to explore and describe the perceptions of delayed antibiotic prescription for respiratory tract infections among general practitioners in Malta.

### Methods

This qualitative phenomenographic study was conducted in Malta. A semi-structured interview guide was developed in English, pilot tested and revised accordingly. Interview topics included views on antibiotic resistance, antibiotic use and delayed antibiotic prescription for respiratory tract infections, and barriers and facilitators to antibiotic prescription. Individual, face-to-face interviews were held in 2014 with a quota sample of 20 general practitioners and transcribed verbatim. Data were subsequently analysed using a phenomenographic approach.

### Findings

General practitioners perceived delayed antibiotic prescription in five qualitatively different ways: (A) "The Service Provider"–maintaining a good general practitioner-patient relationship to retain patients and avoid doctor-shopping, (B) "The Uncertainty Avoider"–reaching a compromise and providing treatment just in case, (C) "The Comforter"–providing the patient comfort and reassurance, (D) "The Conscientious Practitioner"–empowering and educating patients, and limiting antibiotic use, and (E) "The Holder of Professional Power"–retaining

publicly because some information disclosed by general practitioners may make them identifiable, particularly since the study has been carried out in a small country. Providing full transcripts may compromise their identity and that of others, going against ethical considerations. All relevant minimal data have however been made unidentifiable and are presented in the manuscript. Select parts of the transcripts may be made available to readers upon request. Kindly contact Vijaylakshmi Prabhu (vijaylakshmi.prabhu@ki.se) for any future data requests.

**Funding:** This work was supported by Karolinska Institutet funding for doctoral education (KID-funding 3–1233/2013). It was also supported by funding available to CSL at Karolinska Institutet. This study received no other specific grant from any public, commercial or not-for-profit funding agencies. The funders had no role in study design, data collection and analysis, decision to publish, or preparation of the manuscript.

**Competing interests:** At the time of the study, SRK was employed at Karolinska Institutet, Sweden. She is currently employed by the European Centre for Disease Prevention and Control (ECDC). The views and opinions expressed herein are the authors' own and do not necessarily state or reflect those of ECDC. ECDC is not responsible for the data and information collation and analysis and cannot be held liable for conclusions or opinions drawn. This does not alter our adherence to PLOS ONE policies on sharing data and materials. EASG, MR, MAB and CSL have declared no competing interests exist.

general practitioner responsibility by employing a wait-and-see approach. Although general practitioners were largely positive towards delayed antibiotic prescription, not all supported the strategy; some preferred a wait-and-see approach with follow-up. Many delayed antibiotic prescription users selectively practiced delayed prescription with patients they trusted or who they believed had a certain level of knowledge and understanding. They also preferred a patient-led approach with a one to three day delay; post-dating delayed antibiotic prescriptions was uncommon.

## Conclusions

In this study we have shown that general practitioners hold varying perceptions about delayed antibiotic prescription and that there is variation in the way delayed antibiotic prescription is employed in Malta. Whilst delayed antibiotic prescription is utilised in Malta, not all general practitioners support the strategy, and motivations and practices differ. In high consumption settings, formal and standardised implementation of delayed antibiotic prescription could help curb antibiotic overuse. Diagnosis-specific delayed antibiotic prescription recommendations should also be incorporated into guidelines. Finally, further investigation into patients' and pharmacists' views on delayed antibiotic prescription is required.

## Trial registration number

NCT03218930

## Introduction

Antibiotic resistance (ABR) is an imminent threat and a complex challenge that continues to plague modern medicine. It places immense burden on patient outcomes, healthcare expenditure and society as a whole [1]. In Europe, approximately 33,000 people/year die from infections caused by antibiotic-resistant bacteria [1], with an associated cost of €1.5 billion/year in extra healthcare expenditure and productivity losses [2]. Despite more than a decade of coordinated and comprehensive action plans at national, European and global levels, antibiotic use continues to rise in many parts of the world, in all sectors: human, animals and agriculture [3,4].

Antibiotic prescription in primary care is a major driver of ABR [5] and respiratory tract infections (RTIs), although often self-limiting, account for the largest volumes of antibiotics prescribed [6]. Lack of access to diagnostic testing in primary care contributes to diagnostic uncertainty which may drive antibiotic prescription even when not clearly indicated. However studies show that reducing antibiotic prescription does not lead to significantly higher risk of patient complications [7,8].

A number of studies have suggested several strategies to limit unnecessary antibiotic use, including the distribution of national antibiotic guidelines, communication skills training and introduction of point-of-care rapid diagnostic testing, among others [9]. Delayed antibiotic prescription (DAP) has also been shown to decrease antibiotic use for RTIs with little to no overall impact on duration of illness, complication rates and time to symptom resolution [10–16]. Randomised controlled trials carried out in the UK, Norway and Spain showed a 45–80% reduction in antibiotic use through the use of DAP [10–14,16,17].

It is not uncommon for practitioners to initiate DAP strategies of their own accord without well-established guidelines. This results in varying approaches to DAP, including: (i) providing an antibiotic prescription and instructing patients to use it after a given timeframe if symptoms persist or worsen (patient-led strategy), (ii) post-dating (or forward-dating) the prescription so that patients cannot purchase it before a specified date, (iii) instructing patients to return to collect the prescription at a later date if needed, or (iv) requesting that the patient call the clinic/practitioner to issue a prescription should certain criteria be met [18]. An observational study carried out in thirteen European countries showed that DAP practices for acute cough, as well as duration of delay, varies among countries and appears to be less commonly used in southern European countries [19] where antibiotic consumption rates are highest [20]. Despite these variations, few qualitative studies have explored general practitioners' (GPs') perceptions on DAP, particularly in high antibiotic consumption settings.

Malta is a southern European country with the second highest antibiotic consumption rates in Europe [21]. Indeed, in a European-wide survey conducted in 2018, 42% of respondents reported having consumed antibiotics during the past 12 months (EU-average was 32%) [21]. Almost all (96%) were obtained through a prescription by a medical practitioner and the top two reasons for taking the antibiotic were sore throat (22%) and the flu (14%) [21]. A recent surveillance study confirmed that antibiotic prescription rates for RTIs in primary care are high at 45.7% [22]. Of these, 7.2% were DAPs provided for a number of diagnoses, including pharyngitis, sinusitis and bronchitis but also influenza and the common cold [22]. Although DAPs are provided in Malta [22] and are recommended in the national antibiotic guidelines for pharyngitis, rhinosinusitis and bronchitis [23], little is known about how GPs practice DAP, what motivates them to utilise a DAP strategy and their perceptions on it. Given the complexity behind antibiotic prescribing behaviour and the reported advantages of adopting a DAP strategy, there is a need to develop a deeper understanding of prescribers' perceptions of DAP in this context. We therefore aimed to explore and describe the perceptions of DAP for RTIs among GPs in Malta.

## Methods

This study provided baseline data for the Maltese Antibiotic Stewardship Programme in the Community (MASPIC) project, a quasiexperimental social marketing intervention aiming to improve GPs' antibiotic prescribing behaviour for acute respiratory tract complaints in Malta. A study protocol with a detailed description of the design has been published elsewhere [24]. In brief, the intervention was split into three phases, a pre-intervention, intervention and post-intervention phase. Data from the pre-intervention phase, which included interviews, focus group discussions and antibiotic prescribing surveillance, were used to inform the subsequent multi-faceted social marketing intervention. This article presents pre-intervention results.

### Study design

Given the exploratory nature of this study, a qualitative phenomenographic approach was chosen. Phenomenography is a qualitative research method developed to research learning in higher education but has since been used to understand the different ways by which medical practitioners look upon different diseases, medical practice and ABR [25–30]. It seeks to empirically identify the qualitative diversities of how people experience, conceptualise, perceive or understand a specific phenomenon [31]. A phenomenon is typically perceived by people in a limited number of ways, usually between three and seven [25]. There are two characteristics for each understanding of a phenomenon; the structural and referential aspects [32]. The structural aspect explains the part of the phenomenon that is the individual's *focus* of

attention, or awareness, whilst the referential aspect illustrates the meaning of the phenomenon to the person, or how the *meaning* is created [32]. Therefore the different ways that a phenomenon is understood have distinct variations in focus and meaning [25].

## Study setting

In Malta, primary care is provided by GPs in both the private and public sectors. GPs working in the private sector work mostly from solo practices [33]. They often practice within retail pharmacies or private clinics, and house calls are still in demand and fairly common [34]. The physician's fee is paid directly by the patient; no subsidy or reimbursement applies for private consultations. GPs provide services round-the-clock, so many are essentially on-call 24/7. Many private-sector GPs have strong family bonds with their patients, often treating several generations within a family. As a result, it is not uncommon that GPs maintain close contact with regular clients throughout the course of their illness. Primary care services are also available from governmental (public) healthcentres. These GP clinics are walk-in clinics and consultations are completely free of charge. Patients who attend healthcentres are not registered with a specific doctor and are seen by whoever doctor is present on the day. Antibiotics are prescription-only medicines by law. The prescription is considered invalid 10 days after their date of issue. Antibiotics are purchased, out-of-pocket, from community pharmacies [35]. Although non-prescription use was high the past [35], it has reduced substantially and is now the fourth-lowest in the EU/EEA [21].

## Study participants

In phenomenographic research, data from 20 informants is usually sufficient to reach saturation and elucidate the different ways a phenomenon can be understood, perceived or experienced [30]. Therefore a quota sample of 30 registered GPs were called and invited to participate by the first author (EASG). Participants were selected based on a range of characteristics, namely sex, years of GP practice and location of residence. Of the 30 GPs contacted, four were ineligible as they no longer worked as GPs and six declined participation. Therefore the final sample included 20 registered GPs, 14 of whom were male. Age ranged from 32 to 70 years and GP experience ranged from 7 to 45 years (Table 1). GPs' areas of practice were widespread across all the islands.

**Table 1. General practitioners' demographic characteristics (n = 20).**

| | | Frequency, *n* |
|---|---|---|
| **Sex** | male | 14 |
| | female | 6 |
| **Age (years)** | 30–49 | 2 |
| | 40–49 | 4 |
| | 50–59 | 11 |
| | 60–69 | 2 |
| | 70–79 | 1 |
| **Experience in general practice (years)** | 0–9 | 2 |
| | 10–19 | 3 |
| | 20–29 | 10 |
| | 30–39 | 3 |
| | 40–49 | 2 |
| **Healthcare sector of practice** | public | 4 |
| | private | 14 |
| | both | 2 |

## Interview guide development

A semi-structured interview guide was developed following a literature search and based on the researchers' knowledge of the study setting and subject area. The interview guide was developed in English, pilot tested with six GPs and revised accordingly. Interview topics were broad; they focused on views on ABR, antibiotic use for RTIs, barriers and facilitators to antibiotic prescription. Three questions pertaining specifically to DAP (for RTIs) were also included: (i) have you heard about delayed antibiotic prescribing? (ii) please explain how you practice delayed antibiotic prescribing, and (iii) what are your views on delayed antibiotic prescribing? Additional probing or follow-up questions were asked as appropriate in order to clarify participants' responses.

## Data collection

Individual interviews were conducted in English between August and September 2014 by the first author (EASG), a Maltese registered nurse and public health researcher with no previous relationship to the GPs. Although all GPs understood English, in fact the interview questions were posed in English, some felt more comfortable expressing themselves in Maltese and were allowed to do so without inhibiting the interview flow as the interviewer was fluent in both languages. GPs often responded using a mixture of the two languages.

Participants chose the time and location of the interview, often at the GP's workplace outside patient hours. Some preferred other locations such as their home or cafés. Each face-to-face interview lasted between 25 and 67 minutes (median = 40 minutes). Interviewees were encouraged to think aloud and speak openly about their experiences. Concrete examples were sought after in order to avoid superficial descriptions of how things ought to be. They were allowed to explain their understanding of the phenomenon in depth, without being held back by the interviewer. Data were collected until no new information was gained. Interviews were audio recorded and subsequently transcribed verbatim by EASG, performing Maltese to English translations of responses when necessary. Seldom were a few phrases that were hard to interpret discussed with a licensed Maltese proof-reader.

## Data analysis

Data were analysed iteratively following the phenomenographic approach described by Dahlgren and Fallsberg [36] and is outlined in Table 2. Analysis produced an outcome space, a graphical representation that illustrates how the different conceptions or perceptions relate to each other to form the phenomenon [37]. The outcome space is often presented in a hierarchical order since some perceptions may be more complex or include more than one aspect of the phenomenon. EASG and MR analysed data concurrently whilst bracketing their own preconceptions, i.e. making an active effort to set aside any prior assumptions, knowledge or views on the phenomenon being investigated. MR has a background in dentistry and is a researcher with extensive experience in qualitative research methods, particularly phenomenography. Results were discussed with minor adjustments following comments from the remaining co-authors; MAB–a Maltese microbiologist and infection control specialist with particular interest in human behaviour and culture, SRK–a clinical microbiologist with a general understanding of the participants' cultural context and basic knowledge on qualitative research methods, and CSL–a pharmacist by training with extensive experience in qualitative methodologies, including phenomenography.

## Ethical considerations

Ethical approval was sought from the University of Malta Research Ethics Committee and was deemed exempt. Nonetheless, participants were informed about the study's purpose and asked

**Table 2. A step-by-step outline of the data analysis process.**

| 1 | Familiarisation | Transcripts were read repeatedly and independently to gain an overall impression of the material. |
|---|---|---|
| 2 | Compilation and condensation | Each transcript was analysed separately and the most significant statements where GPs described their thoughts about and experiences with DAP were identified and condensed. Transcripts were read keeping the following questions in mind: "What does this tell me about the way the GP perceives DAP?" "What does DAP *mean* to the GP and what is his/her *focus*?" Subsequently, significant elements of the different DAP conceptions were identified. |
| 3 | Comparison, grouping and preliminary description | Similarities and differences between the significant statements were identified, compared and grouped using preliminary labels. Preliminary descriptions of how DAP was perceived were drawn up and labelled for *each* interview. |
| 4 | Formulation and labelling of different categories of description | The various ways of understanding among *all* interviews were collated and compared with one another, looking for similarities and differences. These were then labelled and described as different categories of description. Preliminary categories of description were compared through discussion between the co-analysts to determine if the descriptions captured the variations in views and experiences reported by the GPs. |
| 5 | Final categories of description and outcome space | Preliminary categories of description were presented to and discussed among all co-authors until a consensus was reached and the final categories of description were established. Finally an outcome space was created. |

for written consent. Participation was voluntary and could be terminated at any time if desired. Confidentiality was guaranteed through appropriate data management routines and by using non-identifiable data when presenting results.

## Findings

Five qualitatively different ways of perceiving DAP were identified. These were labelled and described under five categories of description: (A) "The Service Provider", (B) "The Uncertainty Avoider", (C) "The Comforter", (D) "The Conscientious Practitioner", and (E) "The Holder of Professional Power". Selected interview excerpts are included to support the findings.

### A. "The Service Provider": Maintaining a good GP-patient relationship to retain patients and avoid doctor-shopping

GPs within this category feared losing patients from their practice and therefore wanted to maintain a good GP-patient relationship. Consequently DAP was primarily perceived as strategy to maintain a good GP-patient relationship by ensuring that the patient is satisfied with care and will return. In this perception, the focus is on patients, understanding why they want immediate antibiotics and to avoid a second visit for financial reasons. These GPs did not want to be perceived as avaricious. Therefore, by providing a DAP the GP avoided giving patients the impression that they are asked to return for a follow-up visit for financial gain, which may be viewed upon negatively by certain patients.

> "We know people don't want to come to us twice because they don't want to pay the fee. . . So. . . we tell them. . . don't give your kid an antibiotic or try to avoid the antibiotic. . . I will give you a prescription. . . dated not for today. . . for two days' time, and then it's only valid

*for a week... if you get worse buy it. In that way, the patient doesn't think that we're not giving him an antibiotic so that he comes back to us again to pay us."*

*(GP E)*

*"Don't forget this is not a free clinic. People have to pay. I don't want to be seen that I'm abusing them by telling them to come back in two days' time for a prescription and charging them a second time."*

*(GP M)*

Some GPs in this category also perceived that providing a DAP helps avoid doctor-shopping, i.e. patients visiting several GPs for the same illness episode until they obtain their desired antibiotics. According to these GPs, when patients do not get what they want they go elsewhere to seek a prescription and will ultimately acquire them anyway. Therefore DAP was also perceived as a strategy to prevent doctor-shopping and manage patient expectations.

*"We have no option because otherwise... patients will tend to go to another doctor... who will give them an antibiotic straight off!"*

*(GP E)*

GPs in this category described how they used either a patient-led approach to DAP, whereby the GP issues a prescription with the same date of the consultation and provides verbal or written instruction to the patient regarding under which circumstances they should begin antibiotic therapy, or a post-dated approach with a one to two day delay. Two GPs suspected that post-dating prescriptions does not prevent pharmacists from dispensing antibiotics, rendering the method futile. One GP preferred a patient-led approach but requested patients to call for advice before purchasing the antibiotic, while another used post-dated prescriptions and collaborated with pharmacists, ensuring that they were aware that post-dated prescriptions should not be dispensed prematurely.

## B. "The Uncertainty Avoider": Reaching a compromise and providing treatment just in case

The focus for GPs in this category was to provide antibiotic treatment to avoid uncertainty. They perceived DAP as a way of being careful and liked to play it safe. When in doubt, they wanted to make sure that patients had access to antibiotic treatment if necessary. DAPs were provided to patients where the diagnosis was not clear cut and the GP was unsure whether the patient needs an immediate antibiotic prescription but suspected that antibiotic treatment may be required in the subsequent days. DAP solves the "wait-and-see" dilemma of anomalous clinical presentations. The patient is not ill enough to warrant immediate antibiotics but may deteriorate in the coming days and require treatment, therefore GPs believed that providing a DAP offers a good compromise and ensures that the patient is covered in case s/he gets worse. Making sure that the patient will recover and not be subject to complications provided comfort to the GP.

*"I think it's a good compromise... If I have a case where I think it's not severe enough to warrant antibiotics... I will give them a prescription and tell them that they should start if they develop fever or feel worse after 48 hours... or some of these symptoms or signs may change."*

*(GP H)*

*"Especially if you've been seeing this patient before and you know that he is prone to asthma, that he may be a chronic bronchitic, so I see no harm in issuing a delayed prescription if things get worse."*

*(GP I)*

In this category, only one GP described using post-dated DAPs, otherwise GPs used the present date combined with verbal and written explanation to the patient that the antibiotic should not be purchased unless symptoms change or s/he deteriorates within two to three days. Practitioners were selective with whom they provided DAPs. They emphasised the importance of knowing and trusting the patient. In fact, GPs reported being more likely to provide DAPs to regular clients with whom they have an established relationship. If they considered the patient untrustworthy or did not know the patient well enough, GPs usually opted to follow-up instead.

*"You need to be careful to whom you give a delayed prescription. You need to know the patient. I wouldn't give such a prescription to patients I never met before because the minute they leave the clinic they will go to the pharmacist."*

*(GP I)*

One GP expressed concern that by giving a DAP, the doctor must rely on the patient's judgement not only his own. Overcoming this dilemma was accomplished by open communication with patients, requesting them to call back before purchasing the antibiotic in order to reassess the situation and provide guidance over the phone, seemingly attempting to retain some degree of control and accountability.

*"I would also always insist even if they think they need it [the delayed antibiotic prescription], I would ask them to phone me, to tell me what's happening. At least I would have feedback and then I can make sure that they really need it."*

*(GP J)*

## C. "The Comforter": Providing the patient comfort and reassurance

GPs in this category perceived DAP as a way of being considerate and understanding towards patients who fear that their clinical condition may deteriorate and require antibiotic treatment. Although similar to the "Uncertainty Avoiders" in wanting to play it safe, the rationale of GPs in this category was different. Rather than providing treatment to avoid uncertainty, GPs mainly focussed on being receptive of patients' needs. They empathised with the patient and provided a DAP to provide patient comfort and reassurance knowing that they would have a prescription in hand should they get worse. GPs in this category seemed driven by their emotions and when they perceived that patients were overanxious about their condition, they felt that providing a DAP helps put the patient's mind to rest.

*"If you tell them [patients] I'm not going to prescribe you antibiotics. . . they freak out literally. So, I found that giving them a delayed prescription puts their mind at rest. . . they don't need to come back again if these things happen."*

*(GP S)*

GPs in this category did not provide DAPs to patients unless they had a certain level of understanding and were trustworthy or regular clients. Otherwise they advised the patient to re-consult in person.

*"It depends on whether I see that the patient will follow my orders or not. . . There are some people. . . who you cannot even be certain whether they understood you or not even though you explain things to them over and over again. Then there are others who are eager to understand and to follow it out. . . So to those I have a higher tendency to do that [give a delayed antibiotic prescription]. . . then there are those who are literally too scared and will probably not wait. . . then you're risking."*

*(GP G)*

GPs in this category questioned whether DAP is indeed best practice. To them, a pitfall of DAP is that GPs cannot know whether patients actually follow their advice following receipt of a DAP, which is why trust was a dominating decisive factor for utilising this strategy.

*"I'm not sure about what happens actually at the pharmacy . . . I don't know what they [the patients] are doing at the end of the day. So I'm not 100% sure whether it's the right thing to do."*

*(GP S)*

As with the "Uncertainty Avoiders", some GPs tried to overcome this challenge by requesting patients to call back for advice prior to purchasing the antibiotic. They also provided thorough verbal explanation to the patient and a written note on the prescription to make it clear to the pharmacist that the antibiotic should not be dispensed on the day of the prescription.

*"Sometimes I might write the antibiotic, I tell them wait two days, I tell them if things get worse phone me and I'll guide you by phone whether to buy it or not, and I find that system works well, but I have to do that with I mean, first of all the patient is intelligent, and the patient which I know regularly."*

*(GP D)*

## D. "The Conscientious Practitioner": Empowering and educating patients, and limiting antibiotic use

Recognising the threat of ABR, GPs within this category were very concerned about the rampant use of antibiotics in Malta and the rise in ABR. GPs in this category showed deeper understanding of the concept of DAP and were avid users of this method. They perceived DAP as a useful tool for educating and empowering patients, providing an opportunity to teach and convince them that antibiotics are not always necessary.

*"I think it's [delayed antibiotic prescription] a useful tool, it's a tool especially for educating people as I said, you know they realise that the child got better without the antibiotic. . . it also gives the. . . patient power that he can be in charge. . . I think it's very educational. . . it's a good tool."*

*(GP B)*

DAP was also perceived as a strategy intended to limit antibiotic use when they are not required thereby in their view, practising safer medicine.

*"It's a good practice [delayed antibiotic prescription]... I think you're increasing safety in practice and decreasing use when it's not required especially in children."*

(GP P)

To these GPs, mutual trust was important. In fact, they selectively provided DAPs to patients with full understanding of the concept and who had a certain level of knowledge and awareness. They also reflected upon the impact of DAP on their own practice, on how patients respond positively to this approach and the long-term societal gains derived from it.

*"I do a lot of delayed prescribing when I realise the mother is capable of understanding. When the mother walks in and demands antibiotics I wouldn't do it because I know that she's going to give it."*

(GP B)

One private sector GP mentioned working closely with pharmacists so as to ensure that the DAP strategy works as intended. He also stressed the importance of patient communication in DAP and kept in touch with patients to help guide them should they need to dispense the DAP.

*"...a lot of them come to the pharmacy so there's already this relationship... when I prescribe normally I come out of the clinic, go to the pharmacist, explain what he's going to do."*

(GP P)

*"I'll keep in touch with them, and I've got notes so in that case if they don't phone me up I'll phone them up, at least pass on a message."*

(GP P)

## E. "The Holder of Professional Power": Retaining GP responsibility by employing a wait-and-see approach

In this perception the GP focuses on him/herself and how, as a doctor, s/he should act according to his/her training and expertise, to determine if and when patients need antibiotics. GPs within this category were very sure of themselves regarding when antibiotics should and should not be given. As the medical professional, they believed that ensuring good patient outcomes is their sole responsibility. Therefore antibiotic prescription decisions should not be transferred onto the patient. As a result, GPs with this perception considered DAP unsafe practice. They did not practice DAP and were not interested in utilising a DAP strategy in the future. They would rather use a wait-and-see approach in cases where they do not believe that antibiotics are warranted at the time of the first consultation. To them, it was important and justified to re-evaluate the patient in person during a follow-up visit if symptoms persist or worsen rather than give a prescription.

*"I usually don't do it [delay a prescription]. I usually caution the patient and tell him, look here, if you get a temperature, if the conditions gets worse, just call again. Just to make sure that it's really needed . . . I don't do it on purpose; I think it's justified. . . to revise the situation."*

(GP C)

These GPs also shared the view that patients are generally non-compliant to treatment and still believe that they will not get better without antibiotics. In fact GPs expressed that they would not trust that patients would follow instruction should they provide a DAP.

*". . .usually if you do that [give a delayed prescription] they go and buy them [antibiotics] just to avoid having to wait. . ."*

*(GP C)*

### Outcome space

The outcome space created does not truly reflect a hierarchical structure among the different categories of description (Fig 1). It rather illustrates how various perceptions in the categories of description relate to one another. Here, categories of description are split into DAP users

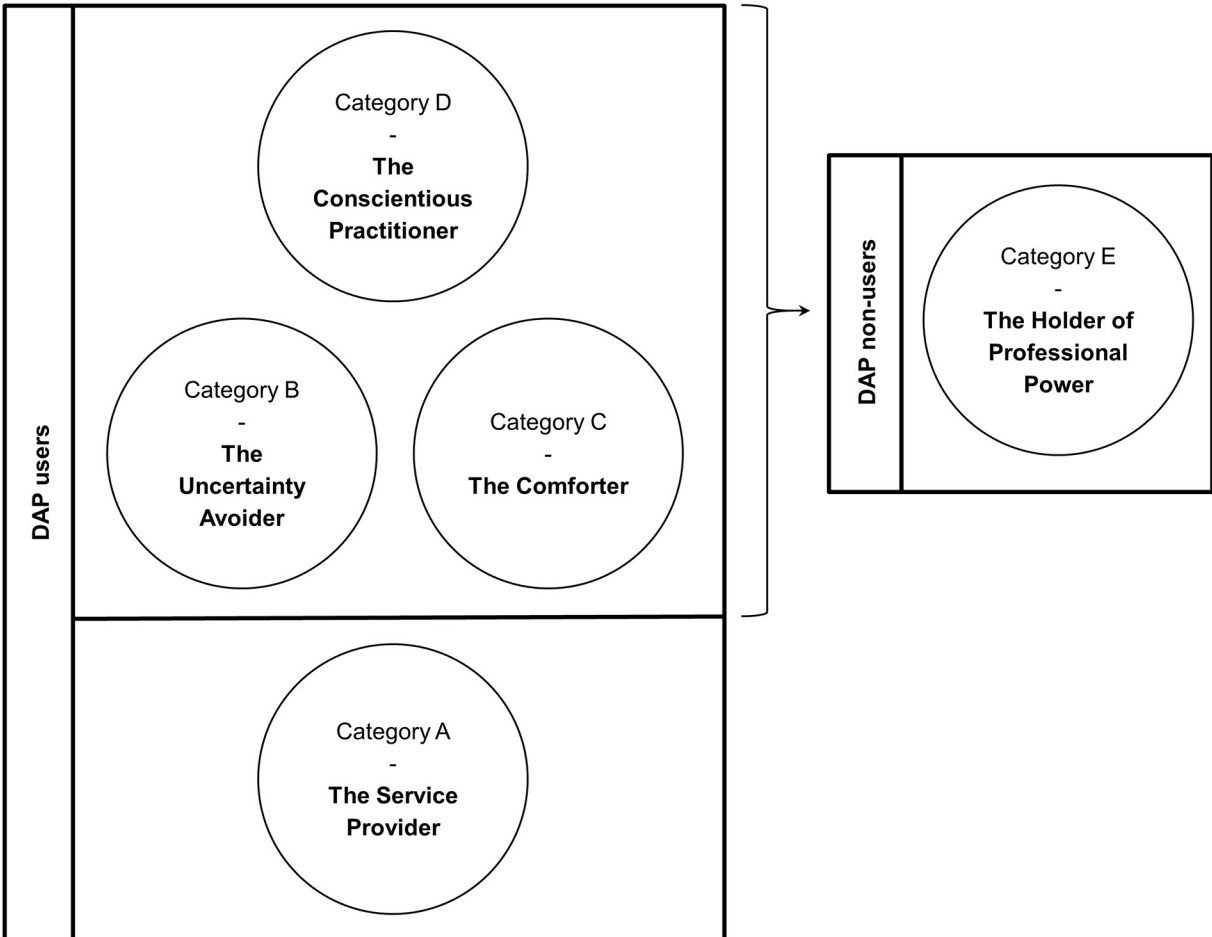

**Fig 1. The outcome space.** This figure illustrates how the categories of description relate to each other. They are split into delayed antibiotic prescription (DAP) users and non-users. "Holders of Professional Power" are unwilling to share decision-making and use a wait-and-see strategy. "Conscientious Practitioners", "Uncertainty Avoiders" and "Comforters" selectively practice DAP; they opt for a wait-and-see approach under certain circumstances, reflecting similarities with "Holders of Professional Power". "Conscientious Practitioners" represent a deeper level of self-reflection and a higher degree of awareness of DAP as a tool for decreasing antibiotic use. "Services Providers" differ; they are non-selective DAPs users who focus on ensuring patient satisfaction and maintaining a good GP-patient relationship, irrespective of who the patient is or what his/her intentions are.

and DAP non-users. "Holders of Professional Power" or non-users, were unwilling to share their decision-making and used a wait-and-see strategy, choosing to follow-up patients whenever necessary. Whilst, "Conscientious Practitioners", "Uncertainty Avoiders" and "Comforters" did practice DAP, they were selective in its use. If they considered a patient untrustworthy, unreliable, lacks certain levels of understanding and awareness, or perhaps they did not know the patient well-enough, then they opted for a wait-and-see approach instead, reflecting similarities with the "Holders of Professional Power". Among this group of DAP users, the "Conscientious Practitioners" represent a deeper level of self-reflection and a higher degree of awareness of DAP as a tool for decreasing antibiotic use. "Services Providers" represent a slightly different group of DAP users who are non-selective when providing DAPs and whose focus is to ensure patient satisfaction and maintain a good GP-patient relationship, irrespective of who the patient is or what his/her intentions are.

## Discussion

### Summary of findings

To our knowledge, this is the first time GPs' perceptions of DAP have been explored in the Mediterranean region. The main contribution of our study is that DAP can be perceived in five qualitatively different ways–(A) to maintain a good GP-patient relationship to retain patients and avoid doctor-shopping, (B) to address clinical uncertainty and reduce the risk of complications, (C) to comfort and reassure the patient, (D) to empower and educate patients, and limit antibiotic use, and (E) not to be practised as prescription is the GPs' responsibility. This corroborates previous results that suggest that antibiotic prescribing is influenced by a range of social and cultural factors [38–42]. While perceptions differed in focus and meaning for the GPs, our findings also revealed how they practiced DAP. A patient-led approach was preferred by many GPs in this study, with GPs sometimes requesting that patients call back before purchasing the antibiotic in order to first receive guidance from the GP. Post-dating DAPs was uncommon and some GPs seemed unfamiliar with the approach. Other modes of delivery, such as collecting the prescription at the clinic at a later date or re-contacting the clinic to request a prescription, were not mentioned by the informants. Instead, a wait-and-see approach with re-assessment of the patient's condition in a follow-up consultation was preferred by some. Although GPs were largely positive towards DAP, not all supported the strategy.

### The GP-patient relationship: Managing expectations and ensuring satisfaction

It has previously been reported that GPs prescribe antibiotics to safeguard their relationship with patients [40] and the same seems to hold true for DAPs although motivations differ [38,43,44]. Several GPs in this study perceived DAP to be a way of ensuring a good GP-patient relationship, thereby retaining patients in their practice. This was largely due to the fact that GPs were concerned that if patients' expectations were not met, dissatisfied patients might choose to visit another GP. Norwegian GPs also reported feeling influenced by patient expectations for antibiotics and would opt for a DAP even if not medically indicated [45], a view shared by UK practitioners [38]. Arroll et al. similarly found that GPs used DAP as a strategy to prevent patients from visiting another GP [43]. Avoiding reconsultation and minimising being perceived as avaricious have been suggested as motivating factors for implementation of DAP [44,46], the latter particularly relevant in fee-for-service healthcare systems. In fact, the same views were expressed by private practitioners in this study where services are financed

through out-of-pocket payments. To these GPs, DAP served as a compromise and ensured patient satisfaction.

It has been shown that patients who expect antibiotics and receive a DAP are more likely to consume them [19,43]. On the other hand, the impact of DAP on patient satisfaction is debatable. A number of randomised trials have found no significant difference in patient satisfaction among various prescription groups [10,11,16]. Conversely there is evidence to show that DAP as compared to immediate prescription could have a negative impact on patient satisfaction [12,13,47]. Patient satisfaction is closely associated to the level of trust in GPs' prescribing decisions and how well patients feel that their concerns have been addressed [16]. Communication techniques that provide GPs with the necessary skills to address patient concerns and expectations without compromising the GP-patient relationship, even if opting for non-antibiotic treatment, could prove beneficial in reducing unnecessary prescription [46].

## Uncertainty avoidance and its implications on DAP

Cultural drivers have arguably provided the best understanding of the heterogeneity of antibiotic use in Europe [48]. These studies have utilised the anthropological model developed by Geert Hofstede, which proposes that national cultures vary along six consistent, fundamental dimensions; individualism, power distance, masculinity, uncertainty avoidance, long-term orientation and indulgence [49]. The dimension which has correlated most consistently with excessive or inappropriate antibiotic use is uncertainty avoidance [50–52]. Countries with high uncertainty avoidance scores do not tolerate uncertainty and ambiguity well; they feel threatened and anxious about it. As a result, physicians may feel urged to do something rather than wait and see [52]. Practitioners in these countries may therefore prescribe antibiotics as a means of seeking reassurance to both the patient and prescriber, even in situations such as colds, flu and sore throat where there is no benefit [51].

Malta has the second highest uncertainty avoidance score in Europe [49]. It is therefore interesting to note the application of DAP in order to avoid uncertainty, especially among "Uncertainty Avoiders" and "Comforters". Whilst "Comforters" perceived DAP as a means of providing patients comfort and reassurance in uncertain situations, "Uncertainty Avoiders" perceived DAP as an appropriate method to ensure that patients have antibiotic treatment options available should their condition deteriorate, even though many RTIs are self-limiting [7]. Studies have also shown that DAP is used by GPs to provide reassurance and comfort to the patient or caregiver [38,42,43,53]. By providing a physical prescription on which a patient can rely should s/he deteriorate, a DAP can be considered a precautionary measure that is better perceived than a non-prescription strategy in situations which do not warrant immediate antibiotic use. Interestingly, studies carried out in the UK and Norway also found that DAP is used as a means to manage diagnostic uncertainty [17,38,42,46], despite being cultures with lower uncertainty avoidance scores than Malta [52]. Nevertheless both of these countries exhibit significantly less antibiotic consumption levels and much lower prescribing for colds, flu and sore throat. DAP could therefore be a potentially useful tool to reduce unnecessary prescribing in high uncertainty avoidance countries, such as those bordering the Mediterranean.

However, DAP does not provide the practitioner with an estimate of the severity of illness. As a result, effective, low-cost, rapid diagnostic testing could play a role in helping reduce antibiotic use. Their introduction could add more diagnostic certainty and therefore eliminate the need for any type of prescription, including DAP. Indeed in an RCT conducted in the Netherlands it was shown that DAPs provided following CRP testing resulted in 50% lower antibiotic use than in the control group [54]. Even so, antibiotics were still prescribed in more than 30%

of acute pharyngitis cases with negative rapid antigen detection testing in Spain [55]. Not surprisingly, this country scores relatively high for uncertainty avoidance.

## Power distance and shifting treatment responsibility onto patients

Several studies have found that prescribers are unwilling to include patients in making a treatment decision and would rather have the patient follow-up in person, deeming it safer practice [38,44,46,53]. Similarly, "Holders of Professional Power" were critical towards DAP because in their perception, it is the GP's responsibility to prescribe antibiotics and therefore providing a DAP is unsafe practice. This consultation style is congruent with a culture that scores high in power distance. Power distance is another Hofstede dimension that describes how hierarchy and power distribution are organised within a society. In countries with high power distance scores, the GP is acknowledged as the expert and may be reluctant to relinquish that expert power; it is one's professional duty to ensure the safety and good health of one's patients. In addition, patients in such societies expect to be told what to do and any hesitation or doubt may be perceived by patients as a sign of lack of confidence [52]. By providing a DAP, GPs may be perceived as indecisive, which is why they might choose to avoid this strategy.

GPs in the Maltese private sector who actively practiced DAP seemed to try to overcome this conundrum by maintaining close contact with patients following a DAP. By doing so they could give advice and instruct patients whether to start antibiotic treatment, thereby retaining some sense of control. Interestingly, studies in the UK found that prescribers felt uncomfortable with the lack of follow-up and not knowing the outcome of the DAP was demotivating [38,46]. Similarly in our study, some DAP users questioned whether DAP is indeed good practice as patients may nonetheless purchase and consume the antibiotics prematurely. Therefore DAP could be perceived as a risky strategy which is why the "Holders of Professional Power" often chose to avoid it altogether.

## DAP as a tool for patient education and empowerment, and to reduce antibiotic use

In contrast to the above, GPs have also expressed that by shifting responsibility, patients are empowered to make their own decisions, giving them an educational opportunity [38,43,45,53]. "Conscientious Practitioners" exhibited a deeper level of insight on the patient benefits of DAP. Not only was it used intentionally to empower and educate patients, but they were also the only GPs who recognised that DAP can help reduce antibiotic use. Both Norwegian and Danish GPs who endorse DAP emphasised that DAP provides an educational opportunity and promotes shared decision-making [45,53]. Danish GPs further viewed DAP as strategy to reduce antibiotic use, a prominent view among New Zealander GPs [43,53].

In stark contrast, UK prescribers did not share these views. They were rather concerned that DAP can provide patients with potentially contradictory messages about when antibiotics should be used [38,46]. British patients also reported receiving confusing messages when issued DAPs after being told that their condition was viral [18]. However studies have nonetheless showed that patients who receive a DAP are less likely to believe that antibiotics are effective for their condition and have lower intentions to re-consult for the same illness in the future [10–12,14,16]. Consequently, although DAP is an effective measure to reduce antibiotic use it is important that DAPs are provided with a clear rationale, explanation and structured advice.

## Delayed antibiotic prescribing in Malta

A recent study showed that most patients with respiratory tract complaints consult GPs within the first three symptomatic days in Malta [22]. Given the lengthy nature of viral respiratory infections, antibiotics are often not recommended for several days, sometimes weeks [56]. Yet all GPs in this study reported providing DAPs with only a one to three day delay. Such a short delay could be explained by the high uncertainty avoidance background. Delaying antibiotics by only a few days in this context might be premature and risks that patients consume unnecessary antibiotics nonetheless. Arguably, under these circumstances, DAPs might indeed provide conflicting messages and patients might incorrectly assume that they require antibiotic treatment despite the fact that they likely have a viral infection [18,38].

It is evident from our study that patient trust, having a well-established GP-patient relationship and the patient's level of understanding and awareness, weighed heavily on a GP's decision whether to provide a DAP or not. Whilst the "Uncertainty Avoiders", "Comforters" and "Conscientious Practitioners" selectively used DAP with patients they believed would follow advice, "Holders of Professional Power" seemed to lack trust that patients will follow advice and in fact preferred to follow-up the patient. Similar selectivity of use has been reported by GPs in Norway, the UK and New Zealand [38,43,45,46].

Some of our informants attempted to overcome these barriers by providing instruction (verbal and/or written), or post-dating the prescription although this was not commonly practiced. Others chose to stay in contact with their patients. In a UK study, GPs were reluctant to post-date prescriptions as they felt they were too restrictive and demonstrated lack of patient trust [38]. These views were not shared by our informants; rather some GPs viewed post-dating as a possible source of confusion for pharmacists who are unfamiliar with the strategy. For doctors who collaborate closely with pharmacists this might not be a problem, but for others this could pose a dilemma and might result in them practising DAP differently, perhaps opting for a more cautious approach.

## Methodological considerations

As is customary in qualitative studies, we used a small, non-probability sample. However, in order to ascertain variation in background factors and gather rich experiences, participants were selected using quota sampling. This strategy considers sizes and proportions of subsamples, with subgroups ultimately reflecting corresponding proportions in the population [57], thus allowing us to illustrate the different ways this phenomenon is perceived in this study setting.

Since participation was voluntary, it is possible that study participants differ from other GPs given that may have been more interested in the topic. Nonetheless, we identified rich variation in GPs' perceptions on DAP. We believe that although our findings are context-specific, they are relevant and could be transferable to other contexts with similar characteristics, especially Mediterranean countries which, like Malta, tend to score high in uncertainty avoidance and power distance.

To ensure trustworthiness of our results we provided a thick description of the research methods used and the findings emerging from the primary data, illustrated by interview excerpts. We also used investigator triangulation by incorporating the perspective of researchers with different professional and cultural backgrounds during the analysis. The research team made an active effort to bracket any preconceptions that might influence the interpretation of the results. A dialogical reliability strategy was applied by reaching a negotiated consensus on the final abstractions summarising our findings. Therefore, the reliability of our study builds upon the interpretative awareness of the research team.

## Conclusions and recommendations for policy, research and practice

In this study we have shown that GPs hold varying perceptions about DAP and that there is variation in the way DAP is employed in Malta. Since DAP is already being practised, better implementation of DAP will not require drastic behaviour changes although several key issues must be addressed. It is evident that requesting that patients return to collect a DAP results in lower antibiotic use than patient-led DAP strategies [10,11,47,58]. A no prescription policy results in even lower antibiotic use [10,47]. In fact it has been argued that DAP could increase the rate of antibiotic use compared to a no prescription policy [58]. Such a policy may be harder to implement than a DAP strategy which could be considered a compromise, addressing patient expectations and concerns, as well as diagnostic uncertainty.

Patient-collection strategies may not be optimal given the Maltese primary care set-up. Instead, post-dating DAPs coupled with structured patient advice and reassurance is probably the safest approach but requires widespread dissemination of information about the purpose of a DAP strategy, formalisation of guidelines and educational opportunities for doctors, pharmacists and the general public alike. Since the effective implementation of DAP entails that pharmacists dispense DAPs appropriately and as intended, further research is needed to better understand pharmacists' views on DAP and their dispensing practices, particularly since some GPs seem to exhibit distrust in pharmacists dispensing DAPs as intended. Awareness and education among the general public must also be improved in order for them to understand the natural course of viral illnesses and that they should engage in self-care before visiting a GP; early consultation should be discouraged. Delayed antibiotic recommendations for each diagnosis that take the natural progression of illness into account should be incorporated into guidelines. For certain conditions like sinusitis and bronchitis it may be worth considering a longer delay.

Since GPs in this study exhibited several and contrasting perceptions of DAP, further investigation into patient views on DAP is required, particularly since GP views on DAP do not always concur with that of patients [43]. Being on the receiving end of the dynamic it is important to understand how patients view DAP in order to put forward recommendations that satisfy both actors in this interaction and to provide GPs with tools to manage various viewpoints.

DAP has been shown to reduce antibiotic use for RTIs, even in high antibiotic consumption settings like Spain [10–16]. We therefore believe that, particularly in high consumption settings like Malta, formal and standardised implementation of DAP could serve as a positive start in the right direction at curbing antibiotic overuse and helping the general public better understand that antibiotics are not always necessary for RTIs.

## Acknowledgments

We would like to express our immense gratitude to all GPs for setting aside time from their busy schedules to participate in this study.

## Author Contributions

**Conceptualization:** Erika A. Saliba-Gustafsson, Michael A. Borg, Senia Rosales-Klintz, Cecilia Stålsby Lundborg.

**Data curation:** Erika A. Saliba-Gustafsson.

**Formal analysis:** Erika A. Saliba-Gustafsson, Marta Röing.

**Funding acquisition:** Erika A. Saliba-Gustafsson, Michael A. Borg, Senia Rosales-Klintz, Cecilia Stålsby Lundborg.

**Investigation:** Erika A. Saliba-Gustafsson.

**Methodology:** Erika A. Saliba-Gustafsson, Michael A. Borg, Senia Rosales-Klintz, Cecilia Stålsby Lundborg.

**Project administration:** Erika A. Saliba-Gustafsson.

**Resources:** Cecilia Stålsby Lundborg.

**Supervision:** Michael A. Borg, Senia Rosales-Klintz, Cecilia Stålsby Lundborg.

**Validation:** Erika A. Saliba-Gustafsson, Marta Röing, Michael A. Borg, Senia Rosales-Klintz, Cecilia Stålsby Lundborg.

**Visualization:** Erika A. Saliba-Gustafsson, Marta Röing, Michael A. Borg, Senia Rosales-Klintz, Cecilia Stålsby Lundborg.

**Writing – original draft:** Erika A. Saliba-Gustafsson.

**Writing – review & editing:** Erika A. Saliba-Gustafsson, Marta Röing, Michael A. Borg, Senia Rosales-Klintz, Cecilia Stålsby Lundborg.

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
