## [Decision Letter · Decision Letter 0]

10 Oct 2019

PONE-D-19-22885

General practitioners’ perceptions of delayed antibiotic prescription: a phenomenographic study

PLOS ONE

Dear Mrs Saliba-Gustafsson,

Thank you for submitting your manuscript to PLOS ONE. After careful consideration, we feel that it has merit but does not fully meet PLOS ONE’s publication criteria as it currently stands. Therefore, we invite you to submit a revised version of the manuscript that addresses the points raised during the review process.

We would appreciate receiving your revised manuscript by Nov 24 2019 11:59PM. To enhance the reproducibility of your results, we recommend that if applicable you deposit your laboratory protocols in protocols.io, where a protocol can be assigned its own identifier (DOI) such that it can be cited independently in the future. For instructions see: http://journals.plos.org/plosone/s/submission-guidelines#loc-laboratory-protocols

We look forward to receiving your revised manuscript.

Kind regards,

Vijayaprasad Gopichandran

Academic Editor

PLOS ONE

Journal Requirements:

1. Please include additional information regarding the interview guide used in the study and ensure that you have provided sufficient details that others could replicate the analyses. For instance, you mention that a pilot guide was tested and altered. On whom was this pilot tested and how many were included?

2. Please remove your figures from within your manuscript file, leaving only the individual TIFF/EPS image files, uploaded separately.  These will be automatically included in the reviewers’ PDF.

Reviewers' comments:

Reviewer's Responses to Questions

**Comments to the Author**

1. Is the manuscript technically sound, and do the data support the conclusions?

Reviewer #1: Partly

Reviewer #2: Yes

2. Has the statistical analysis been performed appropriately and rigorously? 

Reviewer #1: N/A

Reviewer #2: N/A

3. Have the authors made all data underlying the findings in their manuscript fully available?

Reviewer #1: No

Reviewer #2: Yes

4. Is the manuscript presented in an intelligible fashion and written in standard English?

Reviewer #1: Yes

Reviewer #2: Yes

5. Review Comments to the Author

Reviewer #1: This manuscript reports findings of a qualitative phenomenological study that explored the views of GPs in Malta on delayed antibiotic prescription (DAP). The researchers claim that 'this is the first time GPs' perceptions of DAP have been explored in the Mediterranean region'. Funding seems to indicate that this study is part of a doctoral research program.

In general, the reporting in this manuscript seems to have followed the COREQ checklist. I was slightly surprised that the University of Malta HREC has exempted this research from even a minimal risk approval because there are clearly inherent risks of such research with health care professionals - privacy/confidentiality breaches, GPs may feel defensive, GPs may feel they are being scrutinised, GPs may feel obliged to participate, etc. The researchers nevertheless seem to have complied with research ethics protocols. However, whether this is adequate for Plos One, I will leave it with the Editorial Board.

Below are my feedback for the researchers' consideration (particularly those wrt methodological concerns):

1. There is inadequate definition and description of DAP in INTRODUCTION. It is a distinct prescribing practice that needs to be defined properly for non-medical readers.

2. Similarly, there is inadequate description of the MASPIC project that the study is supposed to be providing baseline data for. I assume this is the 1st author's doctoral project??

3. The interviews were conducted in 2014 - 5 years ago?? Why has it taken so long to submit for publication?

4. Description of the 30 GPs contacted and the demographics of the final 20 GPs including Table 1 should be in RESULTS, not METHODS.

5. There is inadequate description of HOW the interview guide was developed. No mention of the questions being translated to Maltese and yet the interviews were conducted in both languages. Presumably then the Maltese interview transcripts would have had to be translated to English as well. Was the translation done by a qualified translator? If not, why not? Was it back-translated to ensure accuracy?

6. Researchers said the three interview questions were open-ended and yet the very first question was a close question. Please explain.

7. The researchers used 'n=' at the end of each of the five themes and these numbers add up to 20, indicating the 20 interviews. I find this totally inappropriate. In fact, throughout DISCUSSION as well as in the last section (outcome space) of RESULTS, the researchers described many areas of overlap of the GPs views across the five categories. So, how can any one interview be so neatly categorised?? I would like the researchers to re-think the way they present their results so that the inherent complexities are not artificially minimised.

8. Discussion was generally well-written although a couple of interesting common features in the five themes were not discussed adequately - eg. GPs seemingly distrust of pharmacists, GPs' views that patients are generally non-compliant.

9. I thought the use of Hofstede's national culture model to help explain the differences in prescribing habits in different countries is appropriate. However, there is inadequate explanation of the dimensions in the model particularly 'uncertainty avoidance' and 'power distance'.

10. Researchers recommend further research to explore patients' views on DAP. Given pharmacists' important role in DAP and yet GPs in this study seemed to have mixed feelings about pharmacists' effectiveness, it will also be necessary to explore pharmacists' views or include pharmacists in future research to ensure the integrity of the implementation of DAP. I think some discussion of this aspect (pharmacists' role) of DAP is also necessary in DISCUSSION.

Reviewer #2: Summary

In their manuscript, “General practitioners perceptions of delayed antibiotic prescription: a phenomenographic study,” Saliba-Gustafsson and colleagues describe a qualitative study of the attitudes of Maltese General Practitioners (GPs) towards delayed antibiotic prescriptions (DAP). The investigators developed an English-language interview guide that covered many topics having to do with antibiotic resistance, antibiotic use, and antibiotic prescribing, including DAP; conducted 20 interviews with GPs; recorded and transcribed the conversations; and had 2 investigators concurrently code the transcripts using a phenomenographic approach.

The investigators divided the 20 respondents into 5 groups based on attitudes towards DAP: 1) Service Providers, 2) Uncertainty Avoiders, 3) Comforters, 4) Conscientious Practitioners, and 5) Power Holders.

Major Comments

The topic is interesting and important. Delayed antibiotic prescribing is used widely, studied often, mentioned in many antibiotic prescribing guidelines, and is controversial. The quotes provided by the authors are illustrative.

To reveal my own bias, I think delayed antibiotic prescribing is a bad idea for a host of reasons. Guidelines are clear about when to use antibiotics for most respiratory tract infections. DAP ignore the natural history of respiratory infections (pharyngitis lasts for 5-7 days, colds 10-14 days, acute bronchitis 3 weeks); does not make microbiological sense; sends a confusing, mixed-message to patients; and does not improve patient outcomes. In using DAP, physicians are abdicating their professional responsibility to patients. Thus, I would categorize myself as a “Power Holder.” Despite being a “Power Holder,” I recognize the potential of “perfection being the worst enemy of the good,” and am open to the possibility that DAP has the potential to reduce overall antibiotic use in the right circumstances.

I have a few General Comments about the conduct of the study and reporting of the manuscript.

First, the interview was much broader than simply asking the GPs about delayed antibiotic prescribing. As such, it is unclear how the coding, analysis, and development of the “graphical space” in the present manuscript relates to other topics covered in the interview guide. How did the investigators decide to focus a manuscript on delayed antibiotic prescribing? Are there other manuscripts that the investigators have or anticipate will emerge from these data?

Second, it is overly simplistic that GPs be categorized into mutually exclusive groups. Individual GPs probably have multiple attitudes and rationales for DAP. Individual GPs could have expressed multiple reasons for DAP. For me personally, as a physician who would probably be categorized here as a “Power Holder,” I like to think that I also practice conscientiously, comfort my patients, and provide good service. Occasionally, I might even strive to avoid uncertainty. Similarly, the GPs in this study did not simply announce themselves in one of these 5 categories. There must have been varying degrees of overlap.

Third, in the Abstract and generally, the authors are never explicit about what clinical situations the GPs are considering when discussing DAP. The three general questions about DAP on page 7, likewise, do not specify a particular clinical situation. Could the GPs have had very different things in mind when discussing DAP? Were they considering only treatment of respiratory infections? Urinary symptoms? GI problems? Other situations?

Minor Comments

Page 3, Line 48: The authors state “This study shows numerous behavioral drivers impact how delayed antibiotic prescription is perceived,” but the term “behavioral drivers” is not defined and is unclear. “Behavioral drivers” sounds like much more specific factors (e.g., environmental cues) that lead to simpler behaviors (e.g., poor diet), rather than the complex socio-cognitive-emotional factors that are leading to the complex behavior of DAP.

Page 4, Line 75: The authors make an excellent argument against DAP. They should make sure that DAP be implemented carefully and conscientiously in the right circumstances if it is principally being used to decrease inappropriate antibiotic use.

Page 5, Line 86: For context, the authors should provide actual metrics of antibiotic use in Malta and comparison countries (high and low utilizing). Other contextual information that would be helpful to the reader would be monthly average Maltese income, the cost of a GP visit, and the out-of-pocket cost of antibiotics.

Page 8, Line 162: Here and elsewhere in the manuscript, the authors need to clarify what they mean by “bracketing their own preconceptions.” It sounds like they are literally using brackets to indicate their own preconceived notions when coding and analyzing the qualitative data. Is that what they mean or is this a more general, figurative term for keeping their own preconceived notions in mind?

Page 10, Line 202: Here and in other places in the manuscript, the authors imply that having the patient call the GP is not an option. In the US, a major reason given for delayed antibiotic prescribing (and antibiotic overprescribing) is efficiency: clinicians do not have time to field phone calls from patients following-up. I was surprised to not see this come up as a reason or a GP archetype (“Efficiency Seekers” maybe).

Page 15, Line 328: The term “Power Holder” has a negative connotation. The 5 GPs who are in this category are doing what they feel, in their best professional opinion, is in their patients’ best interest. A less negative term, like “Professionally Focused” or “Responsibility,” would be better.

Page 18, Line 390: It is not right that “other modes of delivery…were not mentioned by the informants.” There is discussion about returning and calling the clinic.

Page 20, Line 432: Suddenly only referring to the GP categories by letter designations makes the text very hard to follow. The authors should continue using the descriptive categories.

Page 20, Line 433: The authors state that Categories B and C “consistently [applied]” DAP, but it seems that only Category A, “Service Providers,” were consistent in giving out DAPs. The authors need to clarify.

Page 20, Line 449 (as well as Page 4, Line 69): I am skeptical that “low-cost, rapid diagnostic testing” is going to solve antibiotic overprescribing. There is ample evidence of testing being unused or the results being “ignored,” as the authors themselves point out. Most inappropriate antibiotic prescribing is not a result of a lack of diagnostic knowledge – you don’t need more information when someone clinically has a cold – but a result of emotional, social, and cultural factors.

Page 21, Line 454: In describing antibiotic prescribing in the face of a negative rapid antigen test, presumably this was for pharyngitis.

Page 21, Line 461: The authors write “These GPs exhibited a more paternalistic consultation style…” But there were no actual observations of practice in this study. This needs to be reworded such that – if this is true – the GPs “described” a more paternalistic style. (As noted above, I am concerned that the authors are making judgements about the “Power Holders,” that they are not about the other groups.)

6. PLOS authors have the option to publish the peer review history of their article (what does this mean?). If published, this will include your full peer review and any attached files.

Reviewer #1: No

Reviewer #2: No

---

## [Author Response · Author response to Decision Letter 0]

31 Oct 2019

Response to editorial requests:

1. “Please include additional information regarding the interview guide used in the study and ensure that you have provided sufficient details that others could replicate the analyses. For instance, you mention that a pilot guide was tested and altered. On whom was this pilot tested and how many were included?”

- More information regarding the development of the interview guide has now been provided in the methods section (lines 163-165).

2. “Please remove your figures from within your manuscript file, leaving only the individual TIFF/EPS image files, uploaded separately. These will be automatically included in the reviewers’ PDF.”

- Figure has now been removed from the manuscript file as requested and uploaded separately.

Responses to reviewer 1’s comments:

1. “This manuscript reports findings of a qualitative phenomenological study that explored the views of GPs in Malta on delayed antibiotic prescription (DAP).”

- Firstly we would like to thank the reviewer for the thorough comments we received. We have included our comments point-by-point below.

- We would like to clarify that we used a phenomenographic approach rather than a phenomenological approach as they are two different qualitative research methods. Whilst in phenomenology one seeks to explore a phenomenon through people's lived experiences, in phenomenography one investigates the different ways that people understand or perceive a phenomenon, in our case how GPs understand/perceive DAP.

2. “I was slightly surprised that the University of Malta HREC has exempted this research from even a minimal risk approval because there are clearly inherent risks of such research with health care professionals - privacy/confidentiality breaches, GPs may feel defensive, GPs may feel they are being scrutinised, GPs may feel obliged to participate, etc. The researchers nevertheless seem to have complied with research ethics protocols.”

- Despite the fact that following assessment of our study, the research ethics committee in Malta deemed this study exempt of approval (application and letter from the ethics review board can be provided if necessary), we ensured to adhere to ethical guidelines nonetheless. Participants participated voluntarily without coercion or payment. A thorough explanation about the purpose of the study was provided over the phone at the recruitment stage but also in person before the initiation of the interview. Participants were also asked to provide signed written consent. We ensured confidentiality throughout through appropriate data management protocols and by using non-identifiable data when presenting the results.

3. “There is inadequate definition and description of DAP in INTRODUCTION. It is a distinct prescribing practice that needs to be defined properly for non-medical readers.”

- A description of the different ways DAP is employed in practice has now been provided in the introduction as requested (lines 84-90).

4. “Similarly, there is inadequate description of the MASPIC project that the study is supposed to be providing baseline data for. I assume this is the 1st author's doctoral project?”

- A study protocol for the MASPIC project has already been published and is referred to in the text. However, we have now included a very brief outline of the project in order to make it clear that this study pertains to the pre-intervention phase (lines 116-120). You are correct in assuming that this is the first author’s doctoral project. We however we do not think that it is relevant or necessary to note this in the article.

5. “The interviews were conducted in 2014 - 5 years ago?? Why has it taken so long to submit for publication?”

- Due to lack of human resources and the fact that two of the co-authors were out on maternity leave on separate occasions, seeing the project to its completion was prioritised. As a result, work towards a publication was put on hold. Nonetheless, this study shows the pre-intervention results of our project and sets the stage for future publications related to the MASPIC project. Of note, apart from the MASPIC intervention project (which as you correctly said was a doctoral project), no other initiatives in the primary healthcare setting targeting our study population’s antibiotic prescribing behaviour have been carried out, and so we believe that this manuscript it still very relevant. Our future publications will continue to present results of the impact that the intervention had on GPs’ antibiotic prescribing behaviour post-intervention.

6. “Description of the 30 GPs contacted and the demographics of the final 20 GPs including Table 1 should be in RESULTS, not METHODS.”

- It is customary in qualitative phenomenographic studies to present GPs’ characteristics in the methods section. We do not analyse GPs’ demographic data, rather we simply provide it to give the reader a better picture of who our study respondents were. Therefore we strongly believe that describing our study population belongs in the methods section together with a thick description of the study setting.

7. “There is inadequate description of HOW the interview guide was developed. No mention of the questions being translated to Maltese and yet the interviews were conducted in both languages. Presumably then the Maltese interview transcripts would have had to be translated to English as well. Was the translation done by a qualified translator? If not, why not? Was it back-translated to ensure accuracy?”

- More information regarding the development of the interview guide has now been provided in the methods section (lines 163-165).

- Since Malta is a bilingual nation, the interview guides were developed and conducted in English, therefore our description was misleading and this has now been rectified in the test. However, although all GPs understood English, some felt more comfortable responding to the questions in Maltese or often a mixture of both languages, and they were allowed to do so. Since the interviewer was fluent in both languages it did not impact or hinder the interview flow. A better description of this has been provided in the text (lines 173-178).

- The interviews were transcribed verbatim and the responses translated by the first author, who was also the sole interviewer and fluent in both languages. Seldom were a few phrases which were a little harder to interpret discussed with a licensed Maltese proof-reader (lines 185-188).

8. “Researchers said the three interview questions were open-ended and yet the very first question was a close question. Please explain.”

 - Thank you for correctly pointing out that only two of the three questions were in fact open-ended. We have now rectified this in the text (line 167).

9. “The researchers used 'n=' at the end of each of the five themes and these numbers add up to 20, indicating the 20 interviews. I find this totally inappropriate. In fact, throughout DISCUSSION as well as in the last section (outcome space) of RESULTS, the researchers described many areas of overlap of the GPs views across the five categories. So, how can anyone interview be so neatly categorised?? I would like the researchers to re-think the way they present their results so that the inherent complexities are not artificially minimised.”

- Thank you for this important point and observation. This comment is largely reflected in another comment provided later by reviewer 2 (comment #4) and so in line with both comments we realise that providing a frequency to each category of description is indeed very misleading. As is correctly noted, GPs’ perceptions may vary across the various categories of description and GPs can therefore not be categorised in this way. In order to avoid further misinterpretation of our findings, we have removed the number of GPs within each category. Our results should reflect the collectively different ways DAP can be understood.

10. “Discussion was generally well-written although a couple of interesting common features in the five themes were not discussed adequately - eg. GPs seemingly distrust of pharmacists, GPs' views that patients are generally non-compliant.”

- We recommend further research on pharmacists’ involvement in DAP and this has briefly been included in the conclusion (lines 599-602). GPs’ views that patients are generally non-compliant had to do a lot with patient trust; trusting that they will use the DAP as intended. We discuss this briefly in lines 547-553.We also recommend further investigation into patients views on DAP (lines 609-614).

11. “I thought the use of Hofstede's national culture model to help explain the differences in prescribing habits in different countries is appropriate. However, there is inadequate explanation of the dimensions in the model particularly 'uncertainty avoidance' and 'power distance'.”

- We have expanded a little bit more on the different cultural dimensions and the definitions for uncertainty avoidance and power distance in particular (lines 460-463, 465-466, 502-506), and how it impacts GPs antibiotic prescribing as well as patients.

12. “Researchers recommend further research to explore patients' views on DAP. Given pharmacists' important role in DAP and yet GPs in this study seemed to have mixed feelings about pharmacists' effectiveness, it will also be necessary to explore pharmacists' views or include pharmacists in future research to ensure the integrity of the implementation of DAP. I think some discussion of this aspect (pharmacists' role) of DAP is also necessary in DISCUSSION.”

- We appreciate your comment and drawing this very valid point to our attention. Pharmacists’ role in DAP was not something that we focussed on or asked GPs directly about. In fact GPs brought this up spontaneously themselves and so we did not have enough data to delve into this aspect too much. That being said, pharmacists certainly play a very important role in the DAP dynamic and given that some GPs were sceptical towards pharmacists’ practices and that they may be dispensing DAPs prematurely, further investigation is certainly warranted. Since we only briefly discussed this we have expanded a little bit more about it in the text (lines 599-602). In lines 596-599 we briefly mention the need for GPs, pharmacists as well as patients to be included in future development of DAP strategies.

Responses to reviewer 2’s comments:

1. “In their manuscript, “General practitioners perceptions of delayed antibiotic prescription: a phenomenographic study,” Saliba-Gustafsson and colleagues describe a qualitative study of the attitudes of Maltese General Practitioners (GPs) towards delayed antibiotic prescriptions (DAP).”

- Firstly we would like to thank the reviewer for the thorough comments we received. We have included our comments point-by-point below.

- We would just like to clarify that it is not GPs’ attitudes towards DAP that we investigated but rather their perceptions. Describing how one perceives a phenomenon essentially means describing how one views, understands or interprets something and is often ingrained in how one experiences the phenomenon. Investigating a person’s attitudes on the other hand takes it a step further. An attitude is more closely related to the behaviour and explores the person’s stance on the matter, i.e. whether they tend to be in favour of it or not. Perceptions are often considered precursors attitudes.

- Exploring perceptions (rather than attitudes) therefore justifies our use of a phenomenographic study design. In phenomenography, the goal is to describe the different ways that people understand or perceive a phenomenon, in our case how GPs understand/perceive DAP. Rather than exploring attitudes, one explores what a phenomenon (DAP in our case) means to a person and what their focus is. A phenomenon is typically understood by people in a limited number of ways (usually between 3 and 7) and with distinct variations in focus and meaning.

- The study design sub-heading in the methods section has been restructured and we have expanded on phenomenography a little further to better explain the design (lines 123-133).

2. “To reveal my own bias, I think delayed antibiotic prescribing is a bad idea for a host of reasons. Guidelines are clear about when to use antibiotics for most respiratory tract infections. DAP ignore the natural history of respiratory infections (pharyngitis lasts for 5-7 days, colds 10-14 days, acute bronchitis 3 weeks); does not make microbiological sense; sends a confusing, mixed-message to patients; and does not improve patient outcomes. In using DAP, physicians are abdicating their professional responsibility to patients. Thus, I would categorize myself as a “Power Holder.” Despite being a “Power Holder,” I recognize the potential of “perfection being the worst enemy of the good,” and am open to the possibility that DAP has the potential to reduce overall antibiotic use in the right circumstances.”

- We can understand your argument against the use of a DAP strategy to reduce antibiotic use since guidelines are clear that antibiotics can be avoided in many RTI. However cultural perceptions of medical care vary greatly by setting. In settings like ours, guidelines are not necessarily relied on, adhered to, trusted or used, and so we must devise alternative strategies to mediate the problem. Whilst improving guideline adherence is of course a priority, there is plenty of evidence, especially in Europe, that DAP can play an important role in reducing unnecessary antibiotic use, particularly in settings with high uncertainty avoidance cultures where ambiguity and uncertainty are not tolerated well. Numerous studies have shown that DAP can provide the needed reassurance (to both GPs and patients) and can indeed effectively reduce antibiotic use. In addition, although controversial, DAP can have an indirect impact on patients belief that antibiotics are necessary for RTIs and be less likely to re-consult for a similar illness in the future. We hope that this came across clearly in our introduction and we believe that we provided a balanced argument in both our discussion and conclusion.

- That being said, we believe that a more structured, formal and standardised DAP strategy must be implemented to ensure that DAPs are, first and foremost not provided prematurely (DAP guidelines and recommendations must be diagnosis-specific and take the natural progression of illness into account), and provided with clear rationale, explanation and structured advice to the patient in order to avoid mixed messages.

3. “The interview was much broader than simply asking the GPs about delayed antibiotic prescribing. As such, it is unclear how the coding, analysis, and development of the “graphical space” in the present manuscript relates to other topics covered in the interview guide. How did the investigators decide to focus a manuscript on delayed antibiotic prescribing? Are there other manuscripts that the investigators have or anticipate will emerge from these data?”

- As is described in Table 2, although the entire interview transcript was considered in the analysis, particularly since GPs sometimes spontaneously brought up DAP before the interviewer even got to ask any specific questions, most of the data extracted from the interviews were taken from the three questions specified in the methods section. However reading the entire transcript was important; it not only allowed the co-analysts to gain familiarity with the material, but also understand what DAP means to the GP and what their main focus is. As is stated in Table 2, the co-analysts really honed in on significant statements where GPs described their thoughts about and experiences with DAP, whilst keeping the following questions in mind: “What does this tell me about the way the GP perceives DAP?” “What does DAP mean to the GP and what is his/her main focus?”

- As you rightly point out, we are currently working on another article which utilises the rest of the material collected from these interviews. The reason we focused on DAP specifically in this article is that we had very rich data on DAP which allowed us to analyse this particular prescription strategy in more detail. Furthermore, since DAP was later included as an intervention component when the intervention was implemented between 2016 and 2017, exploring GPs’ perceptions on DAP in further depth was important in order to deliver the most effective DAP strategy possible.

4. “It is overly simplistic that GPs be categorized into mutually exclusive groups. Individual GPs probably have multiple attitudes and rationales for DAP. Individual GPs could have expressed multiple reasons for DAP. For me personally, as a physician who would probably be categorized here as a “Power Holder,” I like to think that I also practice conscientiously, comfort my patients, and provide good service. Occasionally, I might even strive to avoid uncertainty. Similarly, the GPs in this study did not simply announce themselves in one of these 5 categories. There must have been varying degrees of overlap.”

- Thank you for pointing this out to us. We would just like to refer you to our methods section where we explained that in phenomenography, “The outcome space is often presented in a hierarchical order since some perceptions may be more complex or include more than one aspect of the phenomenon.” (lines 193-195) In phenomenography it is normal for categories of description to build upon one another. Although our outcome space does not reflect a true hierarchy because we did not identify one, we do understand that GPs do not exclusively fall within distinct categories of description and that by providing numbers we may have misled readers into believing that GPs are categorised into mutually exclusive groups. Whilst the categories of description themselves are mutually exclusive (that is what one strives to identify in phenomenographic studies), as you correctly pointed out, GPs cannot be “placed in boxes” and simply assigned to a category as perceptions may vary across the different categories of description. This is reflected in our description of the outcome space (lines 387-401). Therefore in order to avoid further misinterpretation of our findings, we have now removed the number of GPs within each category. Our results should rather reflect the collectively different ways DAP can be perceived.

5. “In the Abstract and generally, the authors are never explicit about what clinical situations the GPs are considering when discussing DAP. The three general questions about DAP on page 7, likewise, do not specify a particular clinical situation. Could the GPs have had very different things in mind when discussing DAP? Were they considering only treatment of respiratory infections? Urinary symptoms? GI problems? Other situations?”

- GPs were indeed asked to share their views on DAP in the context of RTIs. In order to make this more clear we have now added this to the manuscript title, aim (lines 27-29 and 110-111), as well as the methods section (lines 32-33 and 167).

6. “Page 3, Line 48: The authors state “This study shows that numerous behavioural drivers impact how delayed antibiotic prescription is perceived.” but the term “behavioural drivers” is not defined and is unclear. “Behavioural drivers” sounds like much more specific factors (e.g., environmental cues) that lead to simpler behaviours (e.g., poor diet), rather than the complex socio-cognitive-emotional factors that are leading to the complex behaviour of DAP.”

- This has now been reformulated entirely (lines 50-52).

7. “Page 5, Line 86: For context, the authors should provide actual metrics of antibiotic use in Malta and comparison countries (high and low utilizing). Other contextual information that would be helpful to the reader would be monthly average Maltese income, the cost of a GP visit, and the out-of-pocket cost of antibiotics.”

- We have now restricted the last paragraph in our introduction and included more information on the antibiotic consumption rates in Malta compared to the European average as well as the top two reasons for antibiotic consumption, namely sore throat and the flu (lines 97-103).

- Regarding including the monthly average Maltese income and cost of antibiotics, since we do not discuss this in the paper, we do not believe it to be relevant.

- Since there are currently no official documents stating GPs’ consultation fees it would be hard to provide a number and so we do not feel that we can provide one. As mentioned in the study setting description in the methods section however, public healthcare centres are free-of-charge.

8. “Page 8, Line 162: Here and elsewhere in the manuscript, the authors need to clarify what they mean by “bracketing their own preconceptions.” It sounds like they are literally using brackets to indicate their own preconceived notions when coding and analyzing the qualitative data. Is that what they mean or is this a more general, figurative term for keeping their own preconceived notions in mind?”

- The concept of bracketing is a widely used term in qualitative research. As you have later correctly suggested, what it essentially means is that in order for the researchers’ preconceptions not to influence the findings, one makes an active effort to set aside any prior assumptions, knowledge or views on the phenomenon being investigated. We have added a short explanation in the text (lines 195-197).

- Both the co-analysts possessed different professional backgrounds and preunderstandings of the topic and context. It was therefore very important for us to be aware of our own views and understanding (bracketing) and to discuss our own interpretation of the findings until we finally came to a consensus.

9. “Page 10, Line 202: Here and in other places in the manuscript, the authors imply that having the patient call the GP is not an option. In the US, a major reason given for delayed antibiotic prescribing (and antibiotic overprescribing) is efficiency: clinicians do not have time to field phone calls from patients following-up. I was surprised to not see this come up as a reason or a GP archetype (“Efficiency Seekers” maybe).”

- Using DAP in order to save time was a point never raised by GPs in this context actually. Perhaps it is because of the very close relationship GPs have to their patients as well as the ease of access patients have should they feel that they need to return for a follow up.

- In this manuscript, what we put forward is the fact that we do not believe that patient-collection strategies will be perceived as optimal in this context, primarily due to the fact that some GPs stated that they do not want to be perceived as avaricious by requesting that patients return for follow-up and because GPs often work in single-handed practices on a first-come first-served basis and so from a practical point of view, requesting patients to return simply to collect an antibiotic prescription may be considered burdensome not only by the GP but also the patient.

10. “Page 15, Line 328: The term “Power Holder” has a negative connotation. The 5 GPs who are in this category are doing what they feel, in their best professional opinion, is in their patients’ best interest. A less negative term, like “Professionally Focused” or “Responsibility” would be better.”

- The terms ‘Uncertainty Avoidance’ and ‘Power Distance’ are specific Hofstede socio-anthropological nomenclatures and his dimensions have provided the main explanations for the variation in antibiotic use and quality indicators within EU countries. In this setting, we believe that our term is appropriate and best describes those doctors who are typically reluctant to involve patients in the prescribing decision as they believe that it is their sole responsibly as the healthcare professional. Describing this group as “Power Holders” has no bearing on their quality or conscientiousness of medical practice.

- However, we want to avoid being perceived by readers as judgmental as we do not intend to negatively characterize informants. In fact we believe that we describe “Power Holders” in our results objectively and with slight changes to the discussion based on the reviewers’ suggestions, we have made sure to continue to remain impartial. Nonetheless, we have modified “The Power Holder: retaining GP responsibility by employing a wait-and-see approach” slightly to “Holder of professional power: retaining GP responsibility by employing a wait-and-see approach”.

11. Page 18, Line 390: It is not right that “other modes of delivery…were not mentioned by the informants.” There is discussion about returning and calling the clinic.

- GPs typically request that patients call back to follow-up with the patient, either to (a) provide over-the-phone guidance so that the patient can then go purchase the antibiotic using a prescription that was provided to them during an earlier face-to-face consultation, or to (b) request the patient to return to the clinic for another in-person visit. We have now made this clearer in the text (lines 436-437). As we described later in the text, GPs do not mention requesting patients to come back simply to pick up a prescription as in done in some other settings. As we also mentioned that some GPs may choose not to provide delayed antibiotic prescriptions at all, but rather use a wait-and-see approach, requesting patients to return for a follow-up should they feel worse.

12. “Page 20, Line 432: Suddenly only referring to the GP categories by letter designations makes the text very hard to follow. The authors should continue using the descriptive categories.”

- This has now been modified to include the descriptive category names as recommended throughout the text.

13. Page 20, Line 433: The authors state that Categories B and C “consistently [applied]” DAP, but it seems that only Category A, “Service Providers,” were consistent in giving out DAPs. The authors need to clarify.

- In order to avoid misinterpretation we have removed “consistent” from the text (line 470-472). 

14. “Page 20, Line 449 (as well as Page 4, Line 69): I am sceptical that “low-cost, rapid diagnostic testing” is going to solve antibiotic overprescribing. There is ample evidence of testing being unused or the results being “ignored,” as the authors themselves point out. Most inappropriate antibiotic prescribing is not a result of a lack of diagnostic knowledge – you don’t need more information when someone clinically has a cold – but a result of emotional, social, and cultural factors.”

- Certainly low-cost rapid diagnostic tests cannot be introduced on their own but as part of a multi-faceted intervention which addresses other factors and needs. If marketed as a package then diagnostic testing may, rather than be used for clinical presentations which are more straightforward (like a cold as you mention), be used for cases/particular diagnoses where GPs are truly doubtful of the aetiology and the test will provide more diagnostic certainty. It may also help reassure patients when they do indeed have a viral infection and antibiotics are unnecessary, particularly in cases where patients might demand them. So we still do believe that, in this setting, rapid diagnostics could prove beneficial. However as we noted above, in combination with other initiatives also aimed at reducing antibiotic use. The impact will of course need to be evaluated.

15. Page 21, Line 454: In describing antibiotic prescribing in the face of a negative rapid antigen test, presumably this was for pharyngitis.

- Thank you for pointing this out. We have now specified in the text that it was indeed acute pharyngitis cases that were prescribed antibiotics despite having a negative rapid antigen test (line 493).

16. Page 21, Line 461: The authors write “These GPs exhibited a more paternalistic consultation style…” But there were no actual observations of practice in this study. This needs to be reworded such that – if this is true – the GPs “described” a more paternalistic style. (As noted above, I am concerned that the authors are making judgements about the “Power Holders,” that they are not about the other groups.)

- The entire sentence has been reworded (line 500-501). In order to further explain high power distance societies as objectively as possible without coming across as judgemental, a thicker description of how a high power distance society impacts both doctors and patients has now been provided (lines 516-520).

---

## [Editor Report · Decision Letter 1]

7 Nov 2019

General practitioners’ perceptions of delayed antibiotic prescription for respiratory tract infections: a phenomenographic study

PONE-D-19-22885R1

Dear Dr. Saliba-Gustafsson,

We are pleased to inform you that your manuscript has been judged scientifically suitable for publication and will be formally accepted for publication once it complies with all outstanding technical requirements.

With kind regards,

Vijayaprasad Gopichandran

Academic Editor

PLOS ONE
---

## [Editor Report · Acceptance letter]

15 Nov 2019

PONE-D-19-22885R1 

General practitioners’ perceptions of delayed antibiotic prescription for respiratory tract infections: a phenomenographic study 

Dear Dr. Saliba-Gustafsson:

I am pleased to inform you that your manuscript has been deemed suitable for publication in PLOS ONE. Congratulations! Your manuscript is now with our production department. 

With kind regards,

on behalf of

Dr. Vijayaprasad Gopichandran 

Academic Editor

PLOS ONE